# Comprehensive Analysis of *Rhodomyrtus tomentosa* Chloroplast Genome

**DOI:** 10.3390/plants8040089

**Published:** 2019-04-04

**Authors:** Yuying Huang, Zerui Yang, Song Huang, Wenli An, Jing Li, Xiasheng Zheng

**Affiliations:** 1DNA Barcoding Laboratory for TCM Authentication, Mathematical Engineering Academy of Chinese Medicine, Guangzhou University of Chinese Medicine, Guangzhou 510006, China; 13202078270@163.com (Y.H.); yzr_gzucm1991@163.com (Z.Y.); hsl318@gzucm.edu.cn (S.H.); 18434376623@163.com (W.A.); 2Traditional Chinese Medicine Gynecology Laboratory in Lingnan Medical Research Center, Guangzhou University of Chinese Medicine, Guangzhou 510410, China

**Keywords:** *Rhodomyrtus tomentosa*, chloroplast genome, species identification, phylogenetic analysis

## Abstract

In the last decade, several studies have relied on a small number of plastid genomes to deduce deep phylogenetic relationships in the species-rich Myrtaceae. Nevertheless, the plastome of *Rhodomyrtus tomentosa*, an important representative plant of the Rhodomyrtus (DC.) genera, has not yet been reported yet. Here, we sequenced and analyzed the complete chloroplast (CP) genome of *R. tomentosa*, which is a 156,129-bp-long circular molecule with 37.1% GC content. This CP genome displays a typical quadripartite structure with two inverted repeats (IRa and IRb), of 25,824 bp each, that are separated by a small single copy region (SSC, 18,183 bp) and one large single copy region (LSC, 86,298 bp). The CP genome encodes 129 genes, including 84 protein-coding genes, 37 tRNA genes, eight rRNA genes and three pseudogenes (*ycf1*, *rps19*, *ndhF*). A considerable number of protein-coding genes have a universal ATG start codon, except for *psbL* and *ndhD*. Premature termination codons (PTCs) were found in one protein-coding gene, namely *atpE*, which is rarely reported in the CP genome of plants. Phylogenetic analysis revealed that *R. tomentosa* has a sister relationship with *Eugenia uniflora* and *Psidium guajava*. In conclusion, this study identified unique characteristics of the *R. tomentosa* CP genome providing valuable information for further investigations on species identification and the phylogenetic evolution between *R. tomentosa* and related species.

## 1. Introduction

The family Myrtaceae has over 3000 species distributed predominantly in tropical and subtropical regions of Australia and America [1]. Within this family, *Rhodomyrtus tomentosa*, an evergreen shrub of genera Rhodomyrtus, is commonly found in east and southeast Asia, including southern China, Japan, and Thailand [2]. *R. tomentosa* is an important plant used in traditional Chinese medicine and has a long history of clinical application. Its leaves, fruits and roots have all been used as alternative medicines, with different medicinal efficacies [3]. In addition, its fruits are one of the most popular foods in the wild. The vital medicinal and nutritional properties of *R. tomentosa* have drawn the attention of researchers in recent years [4,5]. The major chemical components of *R. tomentosa* include hydrolytic tannins, phloroglucin, flavonoids, and triterpenes [6], which possess antioxidant, anti-inflammatory, anti-tumor, antibacterial and other biological activities [7,8].

Eukaryotic cells possess large amounts of nuclear DNA, among which there are two organelles that carry independent genetic material, namely mitochondria and chloroplasts (CPs). CPs contain the enzymatic machinery necessary for photosynthesis and other important metabolism, and generally have small, highly conserved genomes [9]. The CP genome, also known as CP DNA, can be duplicated, transcribed and involved in expression [10]. Furthermore, compared to nuclear genomes and mitochondrial genomes, CP genomes are much smaller, and often have a conserved tetrad structure, which facilitates rapid and efficient sequencing and assembly. In addition, the nucleotide evolution rate of the chloroplast genome is moderate. This has allowed chloroplast genomes to be used for phylogenetic studies at different taxonomic levels [11]. The first chloroplast genome was successfully obtained from *Nicotiana tabacum* [12]. Subsequently, the CP genomes of several species, including *Arabidopsis thaliana* [13] and *Panax ginseng* [14], were determined using the first generation of sequencing technology, also called the Sanger method. With significant advances in sequencing technologies and bioinformatics, elucidation and characterization of CP genome has become more rapid and efficient and smoother than ever before [15].

CP genomes are circular DNA molecules generally ranging from 115 to 165 kb in length [16]. A pair of inverted repeat (IR) regions are among the largest components of the CP genome, which are reverse complement to another and are separated by a large-single-copy (LSC) region and a small-single-copy (SSC) region [17]. Two of the most important factors linked to size changes in the CP genome are the expansion and contraction of the IR regions [18]. 

Overall, literatures reports on complete CP genome sequences in the Myrtaceae are still very scarce, hindering phylogenetic analyses based on large-scale genomes. Therefore, it is particularly important to supplement the chloroplast genome information in the Myrtaceae for the phylogenetic purpose. Here, a comprehensive analysis of the complete CP genome of *R. tomentosa* was reported. A detailed map of the CP genome was constructed to help identify characteristics of the *R. tomentosa* CP genome, codon usage and RNA editing sites in *R. tomentosa* CP genome were analyzed to facilitate a better understanding of *R. tomentosa* CP genome. For the purpose of identifying conserved and unique features of CP genomes in different species, four species closely related to *R. tomentosa* were selected for comparison. Interestingly, one protein-coding gene, *atpE* was found to a have a premature termination codon (PTC). Although there are few studies related to PTCs in plant CP genomes, the discovery of PTC in *atpE* in *R. tomentosa* may provide a basis for further studies at the protein level through cloning and expression.

Phylogenetic trees were constructed based on the CP genomes of *R. tomentosa* along with several species within the same family and several unrelated species for the purpose of establishing the evolutionary position of *R. tomentosa*. Our results establish a better understanding of evolutionary history of the Myrtaceae clade and may accelerate phylogenic, population, and genetic engineering research on *R. tomentosa*.

## 2. Results and Discussion

### 2.1. Analysis and Discussion

#### 2.1.1. Characters of *Rhodomyrtus tomentosa* Chloroplast (CP) Genome

The CP genome of *R. tomentosa* displays a typical circular double-chain structure, which is 156,129 bp in length. As shown in Figure 1 and Table 1, it is composed of four parts. A pair of inverted repeats (IRa and IRb) with lengths of 25,824 bp separates the large single-copy (LSC) region from the small single-copy (SSC) region, which have lengths of 86,298 bp and 18,183 bp, respectively. The overall length of the protein-coding sequence (CDS) is 76,113 bp. The total GC and AT contents of the *R. tomentosa* CP genome are 37.1% and 62.8%, respectively. Furthermore, the GC contents of the IR regions, the SSC region and the LSC region are 42.9%, 30.8% and 35.1%, respectively. Generally, the GC content of CP genomes ranges from 34% to 40%, but GC content is not evenly distributed in various regions of the genome. GC content in the IR region was significantly higher than that in the LSC and SSC region. The high GC content in the IR regions was due to the presence of four rRNA genes (*rrn16*, *rrn23*, *rrn4.5* and *rrn5*). The content of GC in CDS (37.6%) is slightly higher than that of total GC content. In greater detail, the GC content for the first, second, and third codon positions are 44.9%, 37.3%, and 30.6%, respectively. Obviously, there is a bias toward a higher AT representation at the third codon position, a feature that was also found in other plant CP genomes [19,20].

The CP genome of *R. tomentosa* encodes a total of 114 different genes (Table 2), of which 15 genes are duplicated in the IR regions. These 129 genes are comprised of 84 protein-coding genes, 37 tRNA genes and eight rRNA genes. Three pseudogenes (*ycf1*, *rps19* and *ndhF*) are located around the IR-SSC, IR-LSC and SSC-IR boundaries, respectively. Four protein-coding genes, seven tRNA genes and four rRNA genes are duplicated in the IR regions. The coding regions constitute 56.7% of the genome, while the rest of the genome contains non-coding regions including introns, pseudogenes, and intergenic spacers.

In the *R.tomentosa* CP genome, there are 18 genes containing introns (Table 3) that may participate in regulating gene expression and enhancing the expression of exogenous genes at specific sites and specific times in the plant [15]. Among those, six are tRNA genes and 12 are protein-coding genes. Most genes contain only one intron, while *ycf3* and *clpP* contain two introns. The *rps12* gene is unusual, containing one 5’ exon and two 3’ exons. The 5’ exon is located in the LSC region, while the 3’ exon is located in the IR regions, which is consistent with the CP genomes of *Psidium guajava*, *Eugenia uniflora* and *Eucalyptus grandis* [21]. The three pseudogenes which contain *ycf1*, *rps19* and *ndhF* are located between IRB/SSC, IRA /LSC and SSC/IRA, respectively. Due to the inverse repeating nature of the IR regions, these three genes cannot be fully duplicated and lose the ability to encode complete proteins, which leads to their classification as pseudogene.

#### 2.1.2. Analysis of Premature Termination Codons in the *R. tomentosa* CP Genome

One protein-coding gene (*atpE*) with a premature termination codon (PTC) was identified during annotation. In order to validate this finding, raw (data) reads were used to conduct mapping on the spliced *R. tomentosa* sequences, followed by Integrative Genomics Viewer (IGV) visual processing to examine variable loci. The mapping rate of the *aptE* locus was found to be higher than 99%, suggesting that this locus was indeed variable and resulted in a PTC. PTCs lead to changes in protein coding. Because CP genomes are relatively conserved, especially within the same family, these plants from the Myrtaceae family were selected as control groups: *Psidium guajava*, *Eugenia uniflora* and *Eucalyptus grandis*. The *atpE* genes from these species were extracted using CLC Sequence Viewer (version 8) and then compared with that of *R. tomentosa*. The comparison results are shown in Figure 2. It can be seen that the premature termination of the *atpE* gene in *R. tomentosa* resulted in the absence of an amino acid compared to the three closely control species.

The *atpE* gene encodes a subunit of the chloroplast ATP synthase complex, which participates in photosynthetic phosphorylation necessary for plant growth [22]. As such, this gene is a critical component of the CP genome. Although literatures reports on PTC in genetics are common, few studies have identified PTCs in plant CP genomes. In genetics, nonsense point mutations often result in the production of nonfunctional proteins, assuming these proteins are properly transcribed and translated [23]. To be more exact, the effect of a nonsense mutation point relies on the proximity of the mutation to the original stop codon, and the degree to which functional subdomains of the protein are affected. Some genetic disorders such as thalassemia result from point-nonsense mutations [24,25,26]. 

With this in mind, the discovery of PTC in the *R. tomentosa atpE* gene may establish a foundation for further studies at the protein level through cloning and expression. Future work on CP transcription and translation are needed to verify the presence and functions of PTCs in *atpE* and potentially other genes.

#### 2.1.3. Identification of Long Repeats (LRs) and Simple Sequence Repeats (SSRs)

Repetitive sequences in CP genome have been a major focus of research. There are an abundance of repeated sequences in the CP genome, which are distributed in intergenetic spacer and intron sequences [27]. Long repeats with length greater than 30 bp, might have functions in promoting chloroplast genome rearrangement and increasing population genetic diversity [28]. In order to verify the above-mentioned functions and obtain a comprehensive understanding of the long repeats within the *R. tomentosa* CP genome, the long repeats in CP genomes from four other species, *Psidium guajava*, *Eugenia uniflora*, *Eucalyptus grandis* and *Melastoma candidum* were selected for comparison according to the ties of consanguinity between species. These three species were used to compare and analyze the conserved and unique characteristics of chloroplast CP genomes between different genera of the same family. *M. candidum*, which belongs to another family within the Myrtiflorae order, is the most closely related among other species whose CP genome sequences are available from NCBI except for the three species of Myrtaceae family. Similarly, *M. candidum* was used to analyze differences between species in different families. The resulting data revealed the repeat structure of these four species, demonstrating that there are 38 (14 forward, 24 palindromic), 31 (14 forward, 15 palindromic, 2 reverse), 33 (16 forward, 16 palindromic, 1 reserve), 30 (16 forward, 14 palindromic) and 49 (22 forward, 19 palindromic, 4 reverse, 4 complement) large repeats (LRs) in *R. tomentosa*, *Psidium guajava*, *Eugenia uniflora*, *Eucalyptus grandis and Melastoma candidum*, respectively. In detail, there is no reverse or complement repeats in *R. tomentosa*, similar to *Eucalyptus grandis*. At the same time, complement repeats exists in *M. candidum*, which is not a member of the Myrtaceae family unlike the other four species. Thus, population genetic diversity is revealed by LRs differences, which is consistent with LRs functional analysis (Figure 3). 

Simple sequence repeats (SSRs) are composed of small repeated sequences of 1 to 6 bp, which are extensively distributed in intergenic regions, intron regions, and even protein-coding regions. High mutation rates in these regions also reflect the genetic diversity [29]. CP SSRs, which are widely used in phylogenetic and population genetic analyses [30], are important sources for developing molecular markers. A total of 282 SSRs were identified in the *R. tomentosa* CP genome and were summarized in Table 4, including 173 mononucleotide, 37 dinucleotide, 63 trinucleotide and nine tetranucleotide repeat units. In addition, 98.8% of the mononucleotide SSRs belongs to the A/T type, which is consistent with previous studies where proportions of polyadenine (polyA) and polythymine (polyT) were higher than those of polycytosine (polyC) and polyguanine (polyG) within CP SSRs in many plants [31].

#### 2.1.4. Codon Usage and RNA Editing Sites

In different organisms, synonymous codons occur at different frequencies—this is called preference [30,32]. As for highly expressed genes, the preference of codons is closely related to the abundance of tRNA. An improved understanding of preference of codons will facilitate further studies on the preference of base composition of DNA sequences, finding optimal codons, and designing expression vectors accordingly to improve the efficiency of protein synthesis [33].

In the *R. tomentosa* CP Genome, all the protein-coding genes were composed of 23,939 codons in sum, among which 2,724 codons (accounting for 11.38%) encode leucine and 286 (1.19%) encode cysteine, respectively. These represent the most and least universal amino acids, respectively, out of the 20 amino acids that can be used for protein biosynthesis by tRNA found in the *R. tomentosa* CP genome. The relative synonymous codon usage (RSCU) value (Figure 4 and Appendix A) increases with the quantity of codons that encode for a specific amino acid. As illustrated, most of the amino acid codons, except for methionine and tryptophan, have preferences. This phenomenon was also found in the CP genomes of other species [10,34].

In addition, RNA editing is a very common phenomenon that occurs in plant CP genomes. The core functions of RNA editing include modifying mutations, correcting and regulating translation [35]. RNA editing sites in the *R. tomentosa* CP genome were predicted based on 35 genes by the predictive RNA editor for plants (PREP) program, among which, a total of 20 genes were analyzed and summarized in Appendix A. In sum, 64 RNA editing sites were identified in the *R. tomentosa* CP genome, in which amino acid conversion from serine to leucine occurred most frequently, while threonine to methionine occurred least often.

#### 2.1.5. Contraction and Expansion of IRs in the *R. tomentosa* CP Genome

As mentioned above, the typical quadripartite structure of the CP genome includes two different single-copy regions and two IR regions [36]. Although the inverted repeat regions (IRa and IRb) are the most conserved regions of the CP genome, contraction and expansion at the borders of the IR regions are hypothesized to explain size differences between CP genomes [37,38]. A comparison between *R. tomentosa* and four other closely related species may explain size differences between their respective CP genomes. 

As presented in Figure 5, the IR/SSC and IR/LSC boundaries of *R. tomentosa* (MK_044696) were compared to those in *Psidium guajava* (NC_033355), *Eugenia uniflora*, (NC_027744), *Eucalyptus grandis* (HM_347959) and *Melastoma candidum* (NC_034716). The length of the IR regions in the five CP genomes showed a modest expansion, ranging from 25,824 to 26,390 bp. The IR regions expanded to partially include *rps19*, *ycf1* and *ndhF*, correspondingly creating truncated *ψrps19*, *ψycf1* and *ψndhF* copies at the junction of IRa/LSC and IRb/SSC and IRa/LSC, respectively. Long *ycf1* pseudogene exists in all species, which has been used to analyze CP genome variation in plants [28,38]. Moreover, it has been reported that the *rps19* gene is one of the most abundant transcripts in the CP genome, which exists in most species except for *Eugenia uniflora* and *Eucalyptus grandis*. The *ndhF* gene, related to photosynthesis, was found to be 67 bp, 112 bp, 209 bp away from the IRb/SSC border in *R. tomentosa*, *P. guajava*, *Eugenia. uniflora*, and *Eucalyptus grandis*, respectively. The *trnH* gene is present at the longest distance (32 bp) from the LSC edge in the *R. tomentosa* CP genome.

#### 2.1.6. Comparative CP Genomic Analysis

The whole CP genome sequence of *R. tomentosa* (MK_044696) was compared to those of *Psidium guajava* (NC_033355), *Eugenia uniflora*, (NC_027744), *Eucalyptus grandis* (HM_347959), and *Melastoma candidum* (NC_034716) using the mVISTA program (Figure 6). By comparison, the two IR regions were less divergent than the LSC and SSC regions, which also occurred in most plants [6,39]. Moreover, it was found that the non-coding region was more variable than the coding region, and the different regions may provide candidate DNA barcodes for future studies. In the coding region, most genes were relatively conserved except for *matK*, *accD*, *ndhF*, *ycf1* and *ycf2*. These divergence hotspot regions of the four plant CP genome sequences provided abundant information for developing molecular markers for phylogenetic analyses and plant identification of Myrtaceae species.

#### 2.1.7. Phylogenetic Analysis of the *R. tomentosa* CP Genome

The availability of a complete CP genome provides us with abundant sequence information that can be used to study the molecular evolution and phylogeny of plants [8,40]. To identify the evolutionary position of *R. tomentosa*, the whole CP genomes of 17 species were used to reconstruct a phylogenetic tree using the maximum likelihood (ML) method, in which four species from Myrtaceae along with 13 species from other families were chosen. Figure 7 shows that most nodes were strongly supported by 100 % bootstrap values (BP). Furthermore, *R. tomentosa* exhibited a sister relationship with two species of *Eugenia uniflora* and *Psidium guajava* and then grouped with *Eucalyptus grandis*. These four species all belong to the Myrtaceae family and were clustered distinctly from other families, which could help reveal the relationship between different families and orders. Nevertheless, node branching of this phylogenetic tree showed high consistency with the angiosperm phylogeny group (APG) IV classification system, which is a modern classification system of angiosperms based on the research of molecular system development. This classification situation differs from that of Flora of China, a series of books that summarize the systematic classification of vascular plants (ferns and seed plants) in China. 

Due to the limited availability of CP genome sequences from Myrtaceae deposited in databases, phylogenetic relationships among Myrtaceae plants based on CP genome sequence can be difficult to determine. Therefore, more data is needed to evaluate phylogenetic relationships of Myrtaceae plants in the future.

## 3. Materials and Methods 

### 3.1. Plant Material, DNA Extraction and Sequencing 

Fresh leaves of *Rhodomyrtus tomentosa* were collected from the Medicinal Botanical Garden of Guangzhou University of Chinese Medicine. Total genomic DNA was extracted from clean leaves using a DNeasy Plant Mini Kit (Qiagen, Hilden, Germany). The extracted genomic DNA was measured in terms of purity and integrity by ultraviolet spectrophotometry and gel electrophoresis. DNA samples with good integrity and purity were submitted for library construction and sequencing using an Illumina Hiseq 2000 Sequencing platform (Illumina Inc., San Diego, CA, USA).

### 3.2. Chloroplast Genome Assembly and Annotation

Trimmomatic (v0.36, Max Planck Institute of Molecular Plant Physiology, Potsdam, Germany) was used to filter and trim low-quality reads. The complete sequence of *Psidium guajava* chloroplast genome was downloaded from NCBI and served as a reference. Based on their coverage and similarity, CP-like reads were extracted and assembled using the Abyss2.0 program to form a complete chloroplast genome sequence. BLASTn was used to conduct self-alignment for locating the precise position of the quadripartite structure. In order to verify the assembly, four regions between the IR regions and the LSC/SSC region were confirmed through PCR amplification. 

The preliminarily gene annotation of the R. tomentosa CP genome was performed using the GeSeq online tool (https://chlorobox.mpimp-golm.mpg.de/geseq.Html) with default parameters [41]. The annotation information was further examined and revised manually using the CLC Sequence Viewer (version 8), which was used to compare the CP genome of *R. tomentosa* and the related species, *Psidium guajava*. Since sequences at both ends of the exon are relatively conserved if genes contain introns, the chloroplast introns can be predicted according to the revised annotation file. The Organellar Genome DRAW (OGDRAW) (v1.2, Max Planck Institute of Molecular Plant Physiology, Potsdam, Germany) [42] was used to construct a detailed map of the CP genome. Finally, the whole CP genome of *R. tomentosa* was deposited into GenBank, with an accession number of MK_044696.2.1.

### 3.3. Genome Structure and Genome Comparison

Distribution of codon usage and GC content were analyzed using the Molecular Evolutionary Genetics Analysis (MEGA 6.06, Tokyo Metropolitan University, Tokyo, Japan) [43]. Thirty-five protein-coding genes of the chloroplast genome of *R. tomentosa* were used to predict potential RNA editing sites using the online program Predictive RNA Editor for Plants (PREP) suite [44], with a cutoff value of 0.8. MISA and REPuter (https://bibiserv.cebitec.uni-Bielefeld.de/session) were used to identify SSRs and LRs in the *R. tomentosa* CP genome [45]. For the purpose of comparison among genomes, the mVISTA program (http://genome. Lbl.gov/vista/index.shtml) was used to align the CP genome of *R. tomentosa* with the CP genomes of *Psidium guajava*, *Eugenia uniflora* and *Eucalyptus grandis* [46]. 

### 3.4. Phylogenetic Analysis

A total of 17 complete CP genome sequences were downloaded from the GenBank (NCBI) database. Nucleotide alignments were subjected to phylogenetic analyses with maximum likelihood (ML) using the GTR + G substitution model, which was selected based on model screening. Bootstrap analysis was conducted with 1000 replicates and TBR branch swapping. In addition, *Cinnamomum camphora* was set as the out-group.

## 4. Conclusions

In conclusion, the complete CP genome of *Rhodomyrtus Tomentosa* was obtained using high throughput sequencing, which is 156,129 bp in length and encodes 129 genes. Further analysis on genome structure and genome characteristics revealed that gene structure and gene content of the *R. tomentosa* CP genome are conserved. The phylogenetic analysis indicated that *R. tomentosa* has a sister relationship with *Eugenia uniflora* and *Psidium guajava*. These results provide valuable information for further investigations on species identification and the evolution of *R. tomentosa* and its related species.

## Figures and Tables

**Figure 1 plants-08-00089-f001:**
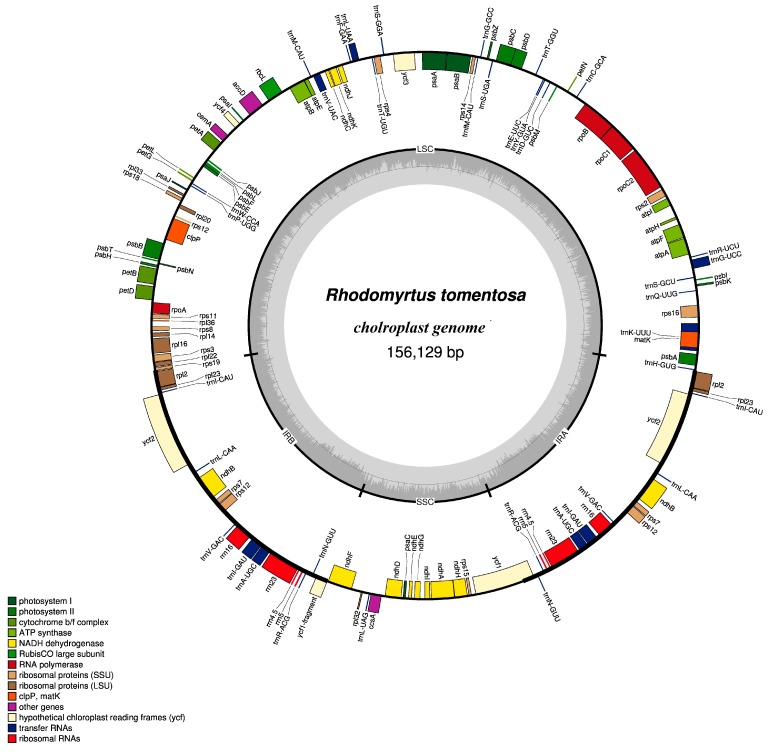
Genome scheme of the *Rhodomyrtus tomentosa* chloroplast genome. Genes inside the circle are transcribed clockwise, while those outside are transcribed counterclockwise Filled colors represent different functional groups that specific genes fall into according to the legend on the bottom. Gray arrow represents gene direction. The darker gray color in the inner circle corresponds to GC (guanine and cytosine) content, whereas the lighter gray corresponds to AT (adenine and uracil) content.

**Figure 2 plants-08-00089-f002:**
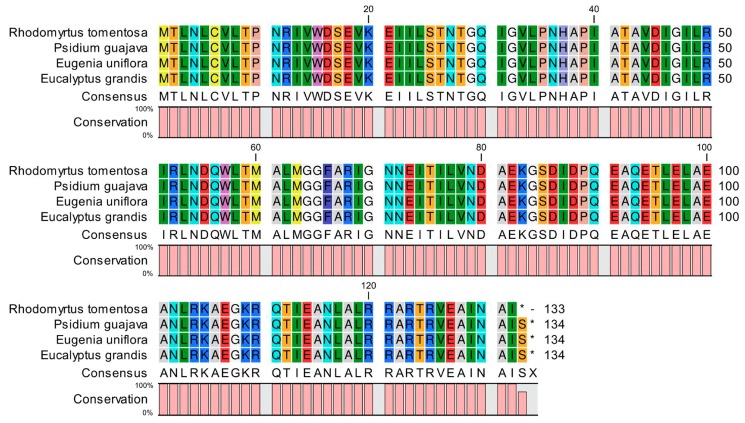
Comparison of amino acid sequence of *atpE* in the *Rhodomyrtus tomentosa* chloroplast genome with those of three closely related species. * Indicates termination codons.

**Figure 3 plants-08-00089-f003:**
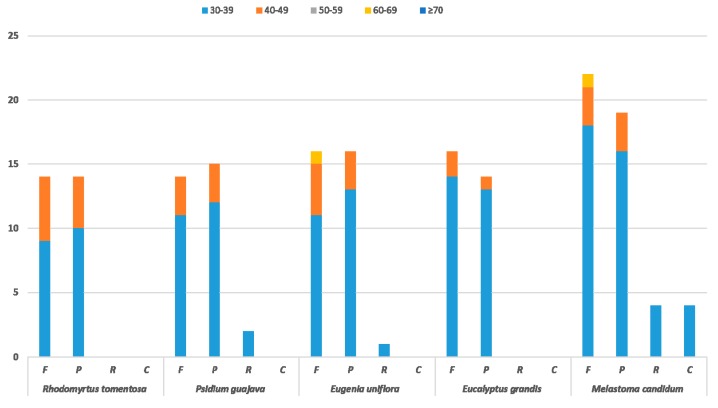
Repeat sequences in four chloroplast genomes. F, P, R, and C indicates the repeat types: F (forward), P (palindrome), R (reverse), and C (complement). Repeats with different lengths are indicated in different colors.

**Figure 4 plants-08-00089-f004:**
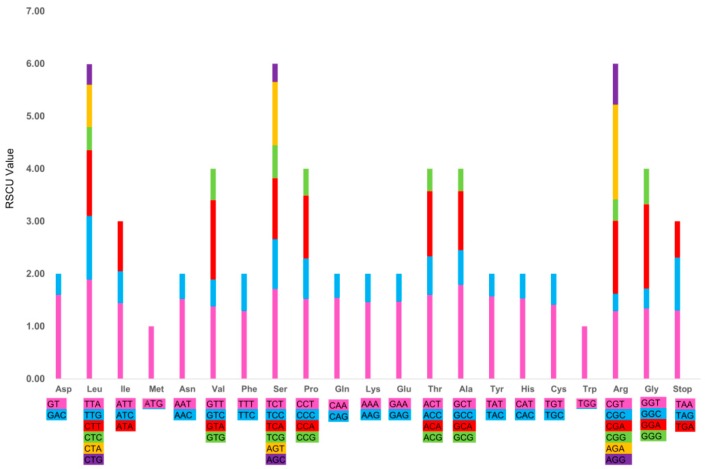
Codon content of 20 amino acid and stop codons in all protein-coding genes of the *Rhodomyrtus tomentosa* chloroplast genome.

**Figure 5 plants-08-00089-f005:**
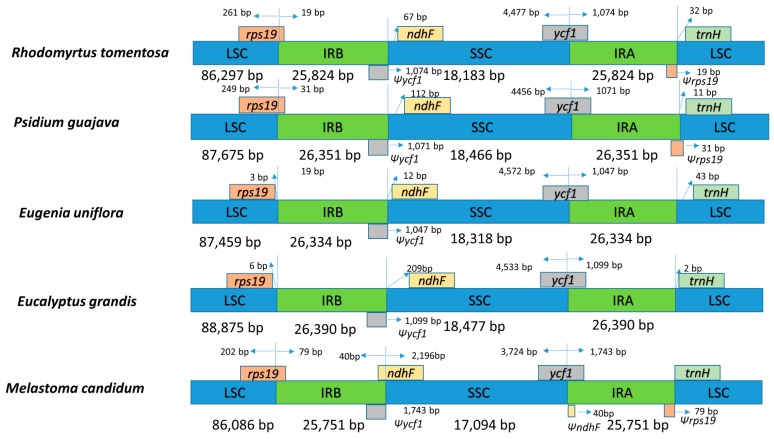
Comparison of the borders of the LSC, SSC, and IR regions among five chloroplast genomes. Ψ: pseudogenes, /: distance from the edge.

**Figure 6 plants-08-00089-f006:**
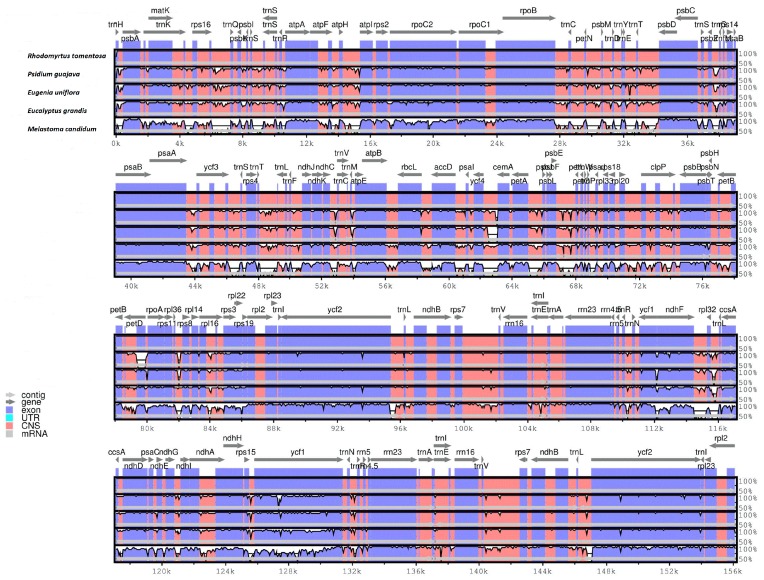
Sequence identity plot comparison of the chloroplast genome of *Rhodomyrtus. tomentosa* with three others using mVISTA. Gray arrows and thick black lines above the alignment indicate genes with their orientation and the position of the IRs, respectively. A cut-off of 70% identity was used for the plots, and the Y-scale represents the percentage identity ranging from 50 to 100%.

**Figure 7 plants-08-00089-f007:**
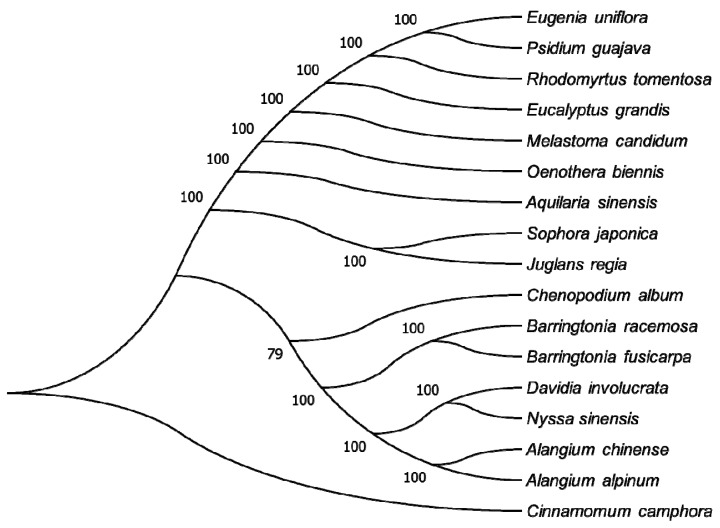
Phylogenetic relationship of the 17 species inferred from maximum likelihood analyses based on the complete chloroplast genome excluding the IRA region. Numbers at nodes represent bootstrap support values.

**Table 1 plants-08-00089-t001:** Base composition in the chloroplast genome of *R. tomentosa*.

Region	Positions	T (%)	C (%)	A (%)	G (%)	Length (bp)
LSC		33.3	18.0	31.7	17.1	86,298
IRA		28.4	20.7	28.7	22.2	25,824
SSC		34.1	16.2	35.1	14.6	18,183
IRB		28.7	22.2	28.4	20.7	25,284
Total		31.8	18.9	31.0	18.2	156,129
CDS		31.6	17.5	30.8	20.1	76,113
	1st position	24.0	18.6	30.8	26.3	25,371
	2nd position	33.0	19.8	29.6	17.5	25,371
	3rd position	37.0	14.1	32.0	16.5	25,371

**Table 2 plants-08-00089-t002:** Base composition in the chloroplast genome of *R. tomentosa*.

Gene Classification	Gene Names	Number
Photosystem I	*psaA*, *psaB*, *psaC*, *psaI*, *psaJ*	5
Photosystem II	*psbA*, *psbB*, *psbC*, *psbD*, *psbE*, *psbF*, *psbH*, *psbI*, *psbJ*, *psbK*, *psbL*, *psbM*, *psbN*, *psbT*, *psbZ*	15
Cytochrome b/f complex	*petA*, *petB* *, *petD* *, *petG*, *petL*, *petN*	6
ATP synthase	*atpA*, *atpB*, *atpE*, *atpF*, *atpH*, *atpI*	6
NADH dehydrogenase	*ndhA* *, *ndhB* * (×2), *ndhC*, *ndhD*, *ndhE*, *ndhF*, *ndhG*, *ndhH*, *ndhI*, *ndhJ*, *ndhK*	12(1)
RuBisCO large subunit	*rbcL*	1
RNA polymerase	*rpoA*, *rpoB*, *rpoC1*, *rpoC2*	4
Ribosomal proteins (SSC)	*rps2*, *rps3*, *rps4*, *rps7* (×2), *rps8*, *rps11*, *rps12* ** (×2), *rps14*, *rps15*, *rps16* *, *rps18*, *rps19*	14(2)
Ribosomal proteins (LSC)	*rpl2* (×2), *rpl14*, *rpl16*, *rpl20*, *rpl22*, *rpl23* (×2), *rpl32*, *rpl33*, *rpl36*	11
Ribosomal RNAs	*rrn 4.5* (×2), *rrn 5* (×2), *rrn 16* (×2), *rrn 23* (×2)	8(4)
Protein of unknown function	*ycf1*, *ycf2* (×2), *ycf3* **, *ycf4*	5(1)
Transfer RNAs	37 tRNAs (8 contain an intron, 7 in the inverted repeats region)	37(7)
Other genes	*accD*, *ccsA*, *cemA*, *clpP*, *matK*	5
Total		129

* Indicates gene contains one intron; ** indicates two introns; (×2) indicates the number of the repeat unit is 2.

**Table 3 plants-08-00089-t003:** Base composition in the chloroplast genome of *R. tomentosa*.

Gene	Location	Exon I (bp)	Intron I (bp)	Exon II (bp)	Intron II (bp)	Exon III (bp)
*atpF*	LSC	144	739	411		
*clpP*	LSC	69	852	296	637	228
*ndhA*	SSC	552	1058	540		
*ndhB*	IR	777	681	756		
*petB*	LSC	6	778	642		
*petD*	LSC	7	750	473		
*rpl16*	LSC	9	988	399		
*rpl2*	IR	391	664	434		
*rpoC1*	LSC	451	733	1619		
*rps12*	LSC	114		232	546	26
*rps16*	LSC	40	860	212		
*trnA-UGC*	IR	35	803	38		
*trnG-UCC*	LSC	23	734	48		
*trnI-GAU*	IR	37	949	35		
*trnK-UUU*	LSC	35	2469	37		
*trnL-UAA*	LSC	35	505	50		
*trnV-UAC*	LSC	37	587	37		
*ycf3*	LSC	126	754	226	723	155

**Table 4 plants-08-00089-t004:** Base composition in the chloroplast genome of *R. tomentosa*.

SSR Type	Repeat Unit	Amount	Ratio (%)
Mono	A/T	171	98.8
	C/G	2	1.2
Di	AC/GT	16	43.2
	AT/AT	21	56.8
Tri	AAC/GTT	6	9.5
	AAG/CTT	18	28.6
	AAT/ATT	24	38.1
	ACC/GGT	2	3.2
	ACT/AGT	3	4.8
	AGC/CTG	5	7.9
	AGG/CCT	1	1.6
	ATC/ATG	4	6.3
Tetra	AAAG/CTTT	2	16.7
	AAAT/ATTT	3	25.0
	AAGC/CTTG	1	8.3
	AAGT/ACTT	1	8.3
	AATT/AATT	2	16.7
	AGAT/ATCT	3	25.0
Penta	ACCGG/CCGGT	2	100.0

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
