# Peer review of "Comprehensive Analysis of Rhodomyrtus tomentosa Chloroplast Genome"

_plants, 2019, doi:10.3390/plants8040089_

Round 1
Reviewer 1 Report
Huang, et al reported the chloroplast genome of Rhodomyrtus tomentosa and analyzed in depth the features of the CP genome. Several differences and unique properties were identified in the R. tomentosa CP genome comparing to other species.
I have two major concerns about this manuscript. The authors show a large amount of information on the R. tomentosa CP genome features by comparing to the other three or four CP genomes. However, I just don’t know why the other species were picked for analysis, are they closely related species, or other reasons? Why sometimes the author picked three, sometimes picked four? What is the purpose of the comparative studies? Secondly, again, why did the authors pick 17 species for the phylogenetic study, what is the purpose of this study? Any interesting found?
With respect to each section, I have the following comments:
Abstract
L28, remove A
L29-L30, Latin names should be in italics
Introduction
The entire section needs to be revisited and expanded.
The authors mentioned several times about the other species including Psidium guajava, Alangium chinense, and Eucalyptus grandis in their analysis. More information is needed for those three species, why did author want to include those three species in their analysis?
Moreover, I don’t see any introduction of the phylogenetic study, why did the author want to include this analysis?
Need more introduction of the topics selected in the paper, for instance, why the three genes were selected for premature stop codon analysis, are they important?
Results and Discussion
L77, Characters or Features of not Characteristics
Fig. 1 has too low resolution, which is not easy to judge each part.
Table 1, why did the authors put the symbol uridine in parenthesis here? As an alternative as thymine? Is uridine present in the CP genome?
L94 the IR regions? Are you talking about two IR regions? Or just one?
L99, Table 3, the top line is disorganized, which make the table difficult to read. Please revise.
L103, in which, not in that
L109, revise the subtitle which is confusing.
In addition, this section describes the three genes that have premature stop codons. The authors need to show the three sequences somewhere, including the potential ancestor sequences and point out the positions of the premature stop codons. The authors also need to introduce the three genes. What are they? Why are the premature stop codons of the genes interesting/important? Otherwise, it is meaningless to discuss the consequence or genetic reason for those truncated sequences.
L116, genetic? Or genetics
L125, this subtitle has a wrong English structure. Recommended subtitle: Identification of long repeats and SSRs
L126 repetitive not repeated?
126-L127. Are authors talking about repetitive sequences in genomes in general or specifically in CP genomes?
L169, Recommended subtitle: Contraction and Expansion of the IR in the R. tomentosa CP Genome
L187, again, as I pointed out earlier, I don’t know why the author picked the other three genomes, please clarified. Related to the subtitle is confusing, genomic analysis usually mean the nuclear genomes.
Figure 6. The authors need to clarify why they selected these 17 species.
L190 Figure 5 has too low resolution, which is hard to read
Methods
In general, the methods were present clearly. Except for the following items,
L268 the accession number MK_044696.2.1 is not present in Genbank
Please describe how the chloroplast introns were predicted.
L268, why are the words “Analysis and Discussion” added here?
Minor issues,
Between words and left or right parenthesis, there should have space.
There is no such word “intergenetic”, but intergenic, meaning between genes.
All Latin names of species should be in italics.
Subtitles should provide/outline the information for each unit, it is not appropriate to use a big title. For example, Comparative Genomic Analysis, or Phylogenetic Analysis, the authors need to specify title which group of species, which genomes they were analyzing.
Author Response
I have two major concerns about this manuscript. The authors show a large amount of information on the R. tomentosa CP genome features by comparing to the other three or four CP genomes. However, I just don’t know why the other species were picked for analysis, are they closely related species, or other reasons? Why sometimes the author picked three, sometimes picked four? What is the purpose of the comparative studies? Secondly, again, why did the authors pick 17 species for the phylogenetic study, what is the purpose of this study? Any interesting found?
1 Thank you for your concerns. To better explain this, I reselected four species for comparison,
They are Psidium guajava, Eugenia uniflora, Eucalyptus grandis, Melastoma candidum, respectively. The previous three species all belong to the same family with R. tomentosa, where Psidium guajava has the closest relation with R. tomentosa, Eugenia uniflora is the next, and then Eucalyptus grandis, these three species can be used to compare and analyze the conserved and specific characteristics of chloroplast genomes between different genera of the same family. Melastoma candidum, not of Myrtaceae family, but of Myrtiflorae order, is the closest among other species whose chloroplast genome sequence is available on the NCBI except for the three species of Myrtaceae family. Similarly, Melastoma candidum can be used to analyze differences between species in different families.
2. Establishment of the phylogenetic tree is generally choose the species from the same family, and then has a specie from different family as an out group. However, the available chloroplast genomes sequence of myrtaceae plants on NCBI were very few. Therefore, I had to choose the plants of the same order. After screening the plants of the same order on NCBI, I found that the species number is still small, so then I found the ones of the same class. Finally, I found 16 species according to their proximity, and chose the one with the farthest kinship as the out group. Finally, 16 species were selected according to their proximity, and the one with the farthest kinship was selected as the out group. Relatedness was first determined according to the classification of Flora of China, nodes branching of the established phylogenetic tree showed a high consistency with the Angiosperm Phylogeny Group (APG) IV classification system, which is a modern classification system of angiosperms based on the research of molecular system development. This classification situation was different from that of Flora of China, a great series of books that summarize the systematic classification of vascular plants (ferns and seed plants) in China. So I think it is an interesting point that deserve discussing.
Abstract
1. L28, remove A,
2. L29-L30, Latin names should be in italics
I am very sorry for such careless problems, which have been modified in L28 and L29-30. Thank you for your careful review
Introduction
1. The entire section needs to be revisited and expanded.
1. The authors mentioned several times about the other species including Psidium guajava, Alangium chinense, and Eucalyptus grandis in their analysis. More information is needed for those three species, why did author want to include those three species in their analysis?
2. Moreover, I don’t see any introduction of the phylogenetic study, why did the author want to include this analysis?
3. Need more introduction of the topics selected in the paper, for instance, why the three genes were selected for premature stop codon analysis, are they important?
Thank you for your advice, I have revised and expanded the introduction. I briefly introduce the selected four reference species, and explained the reason why I choose the four species (L78-L86). And I introduced the topics selected in L69-L89. Then I introduced the phylogenetics and phylogenetic tree, and the meaning of phylogenetic analysis in L90-L98.
Results and Discussion
I am very sorry that there are so many mistakes since that I have not carefully checked, and sorry to cause you a lot of trouble. As for some of your questions and Suggestions. I have carefully explained and revised. Thank you for your professional advice. If the revision is not perfect, I am looking forward to your professional advice again.
1. L77, Characters or Features of not Characteristics
I've replaced Characteristics with Characters in L101. Thank you for your professional advice.
2. Fig. 1 has too low resolution, which is not easy to judge each part.
I have drawn fig. 1 again by the Organellar Genome DRAW(OGDRAW). Its resolution is up to 5000 x 5000, which may be due to the conversion of the manuscript submission system when the manuscript was submitted, resulting in a low resolution.
3. Table 1, why did the authors put the symbol uridine in parenthesis here? As an alternative as thymine? Is uridine present in the CP genome?
During the annotation process, I found that there were Rrna and Trna, so I added uridine in parenthesis, but there was no uridine in the chloroplast genome sequence, so I deleted the character U in parenthesis.
4 L94 the IR regions? Are you talking about two IR regions? Or just one?
I want to talk about two IR regions, I changed it to the plural in L118, sorry for the careless mistakes.
5. L99, Table 3, the top line is disorganized, which make the table difficult to read. Please revise.
I'm sorry that I didn't find it. Thank you for your careful review, I have revised it.
6 .L103, in which, not in that
Thank you for your professional advice, I replaced that with which in L128
7. L109, revise the subtitle which is confusing.
I'm sorry to confuse you, I have revised it.in L135
In addition, this section describes the three genes that have premature stop codons. The authors need to show the three sequences somewhere, including the potential ancestor sequences and point out the positions of the premature stop codons. The authors also need to introduce the three genes. What are they? Why are the premature stop codons of the genes interesting/important? Otherwise, it is meaningless to discuss the consequence or genetic reason for those truncated sequences.
Due to the chloroplast genome is relatively conservative, especially in the same family, we selected three plants from Myrtaceae family as the control groups which are Psidium guajava, Eugenia uniflora, and Eucalyptus grandis, and extracted the three gene with CLC software, then compared them with the genes of R. tomentosa. The comparison results are shown that only atpE has PTC. So I introduce the atpE briefly,and the discovery of this condition in atpE genes may provide a basis for further studies at the protein level through cloning and expressing.Thank you for your advice, I have revised it in L136-L159
8 .L116, genetic? Or genetics
Genetics is more accurate, thank you for your advice. I revised in L150.
9. L125, this subtitle has a wrong English structure. Recommended subtitle: Identification of long repeats and SSRs
Thank you for your recommendation, I have revised it according to your recommendation in L161
10 L126 repetitive not repeated?
I have revised it in L162, thank you for your professional advice.
11. 126-L127. Are authors talking about repetitive sequences in genomes in general or specifically in CP genomes?
I'm sorry I left out the chloroplast to cause your confusion. I have added chloroplast in L162-L163
12. L169, Recommended subtitle: Contraction and Expansion of the IR in the R. tomentosa CP Genome
Thank you for your recommendation, I have revised it according to your recommendation in L208
13. L187, again, as I pointed out earlier, I don’t know why the author picked the other three genomes, please clarified. Related to the subtitle is confusing, genomic analysis usually mean the nuclear genomes.
I am sorry for confusing you, to better clarify this, I reselected four species for comparison,
They are Psidium guajava, Eugenia uniflora, Eucalyptus grandis, Melastoma candidum, respectively. The previous three species all belong to the same family with R. tomentosa, where Psidium guajava has the closest relation with R. tomentosa, Eugenia uniflora is the next, and then Eucalyptus grandis, these three species can be used to compare and analyze the conserved and specific characteristics of chloroplast genomes between different genera of the same family. Melastoma candidum, not of Myrtaceae family, but of Myrtiflorae order, is the closest among other species whose chloroplast genome sequence is available on the NCBI except for the three species of Myrtaceae family. Similarly, Melastoma candidum can be used to analyze differences between species in different families.
And the wrong words related to the subtitle were deleted by me in L227, thank you for your kind attention. I have replaced the genomic analysis with chloroplast genomic analysis in L227
14 Figure 6. The authors need to clarify why they selected these 17 species.
Establishment of the phylogenetic tree is generally choose the species from the same family, and then has a specie from different family as an out group. However, the available chloroplast genomes sequence of myrtaceae plants on NCBI were very few. Therefore, I had to choose the plants of the same order. After screening the plants of the same order on NCBI, I found that the species number is still small, so then I found the ones of the same class. Finally, I found 16 species according to their proximity, and chose the one with the farthest kinship as the out group. Finally, 16 species were selected according to their proximity, and the one with the farthest kinship was selected as the out group. Relatedness was first determined according to the classification of Flora of China, nodes branching of the established phylogenetic tree showed a high consistency with the Angiosperm Phylogeny Group (APG) IV classification system, which is a modern classification system of angiosperms based on the research of molecular system development. This classification situation was different from that of Flora of China, a great series of books that summarize the systematic classification of vascular plants (ferns and seed plants) in China. So I think it is an interesting point that deserve discussing.
15 L190 Figure 5 has too low resolution, which is hard to read
I have uploaded the picture from the vista program again, Its resolution is up to 3000 x 3000, which may be due to the conversion of the manuscript submission system when the manuscript was submitted, resulting in a low resolution.
Methods
In general, the methods were present clearly. Except for the following items,
1. L268 the accession number MK_044696.2.1 is not present in Genbank
When it was uploaded to Genbank, its staff informed me by email that the uploaded file would be checked. They gave me accession number r first and then released it after confirming that there was no problem. Therefore, it may take some time. I am sorry for it.
Please describe how the chloroplast introns were predicted.
The annotation information was further examined and revised manually by the CLC Sequence Viewer (version 8) which was used to compare the CP genome of R. tomentosa and the related specie Psidium guajava. Since sequence at both ends of the exon is relatively conservative if genes contain introns, the chloroplast introns can be predicted according to the revised annotation file. I have added the describe in L310 to L313
L268, why are the words “Analysis and Discussion” added here?
I am sorry to forget to delete the words “Analysis and Discussion” I revised in L 316. Thank you for your kind attention.
Minor issues,
1.Between words and left or right parenthesis, there should have space.
I am sorry for the wrong formats, and I have revised it.
2. There is no such word “intergenetic”, but intergenic, meaning between genes.
I am sorry for my poor English. Thank you for your professional advice. I have revised it.
3. All Latin names of species should be in italics.
I am sorry again for the wrong formats, and I have revised it. Thank you for your professional advice.
4. Subtitles should provide/outline the information for each unit, it is not appropriate to use a big title. For example, Comparative Genomic Analysis, or Phylogenetic Analysis, the authors need to specify title which group of species, which genomes they were analyzing.
I am sorry for using the big subtitles, thank you for your kind advice, I have replaced Comparative Genomic Analysis and Phylogenetic Analysis with Comparative Chloroplast Genomic Analysis and Phylogenetic Analysis of R. tomentosa.
Reviewer 2 Report
Manuscript submitted by Huang et. al., represents comprehensive analysis of Rhodomyrtus tomentosa chloroplast genome. Overall manuscript written very well, methods described are clear. However, few minor comments that are given bellow need to be addressed
1. Chloroplast DNA was not isolated; instead genomic DNA was sequenced?. please explain this
2. Explain the rationale behind using Psidium guajava chloroplast genome as reference
3. Figure 2 can be modified, follow the format as in figure 3.
4. Citations are irregular follow the order and journal format
5. Line 268 : remove "Analysis and Discussion"
6. This is reference based assembly from different species, so explain the rational behind calling complete Chloroplast Genome
Author Response
1. Chloroplast DNA was not isolated; instead genomic DNA was sequenced?. please explain this
I'm sorry for the confusion. Because there are generally two strategies for studying chloroplast genome. One of them is to isolate the chloroplast genome DNA as you mentioned, then sequence and assemble it. The other is to extract the total genomic DNA of a species, directly carry out high-throughput sequencing, and obtain the chloroplast genome similar sequences by comparison with reference genomes, and then further assemble the chloroplast genome.The first method is more conventional but difficult to extract. The second applies to species with reference genomes. We chose the second method because we found the reference species that were more closely related.
2. Explain the rationale behind using Psidium guajava chloroplast genome as reference
Psidium guajava is the closest relative to Rhodomyrtus tomentosa in the same family which has been reported. Based on the conserved nature of chloroplast genome, Rhodomyrtus tomentosa CP genome can be well annotated by selecting related species as reference.
3. Figure 2 can be modified, follow the format as in figure 3.
Thank for your professional advice, I have modified the figure 2 according to the format of figure 3.
4. Citations are irregular follow the order and journal format
I am sorry for irregularly following the order and journal format, I have revised it.
5. Line 268 : remove "Analysis and Discussion"
I am sorry for not removing the "Analysis and Discussion", thank you very much for your kind reminder.
6. This is reference based assembly from different species, so explain the rational behind calling complete Chloroplast Genome
CP genome exhibits a small size with a conserved structure of tetrad.Due to the strong conservation among the related species, the assembly was conducted with the related species as the reference, The spliced sequence has a redundant segment that overlaps with the LSC region.When the redundancy is removed, a complete chloroplast genome is obtained displays a typical quadripartite structure using the Abyss2.0 program . BLASTn was used to conduct self-alignment for locating the precise position of the quadripartite structure. In order to verify the assembly, four regions between the IR regions and the LSC/SSC region were confirmed through PCR amplification.
Reviewer 3 Report
The presented manuscript covers the sequence-based characterisation and analysis of the complete chloroplast 20 (CP) genome of R. tomentose. Authors have identified the unique characteristics of the R. tomentosa 30 CP genome providing valuable information for further investigations on species identification and 31 the phylogeny evolution between R. tomentosa and its related species. Based on the valuable information obtained through this work, the manuscript is worth considering for publication. I would recommend the authors to review the article to carry out minor typos.
Author Response
Thank you very much for your affirmation and recommendation. I am very sorry for the minor typos in the article. After careful review of the article, I have revised it.
Reviewer 4 Report
Point #1
In general the manuscript is well written in terms of language and organization . Just double check the manuscript for minor spelling. The punctuation and spacing (Ex. P3Line#105 ---- The two pseudogenes , should be; instead---
Point #2
Avoid using the personal pronouns as I, We...(in P3 Line 110 We found three ----
Author Response
Point #1
In general the manuscript is well written in terms of language and organization . Just double check the manuscript for minor spelling. The punctuation and spacing (Ex. P3Line#105 ---- The two pseudogenes , should be; instead---
I am sorry for the minor spelling, I have checked the manuscript twice to avoid it. And thank you for your affirmation.
Avoid using the personal pronouns as I, We...(in P3 Line 110 We found three ----
Thank you very much four your professional advice, I have revised the personal pronounce and try to use the passive voice in Line 134.
Round 2
Reviewer 1 Report
I almost accept the revised manuscript for the publication.
However, I think the authors still need to further revise the Introduction section. For instance, L69-89, the authors put too many details here, some of them should be put in the Conclusion or Discussion sections. L90-93, this is an unnecessary introduction.
In addition, I recommend the authors to further correct and polish their manuscript, as there are still too many typos and inappropriate English expressions.
Author Response
1. However, I think the authors still need to further revise the Introduction section. For instance, L69-89, the authors put too many details here, some of them should be put in the Conclusion or Discussion sections. L90-93, this is an unnecessary introduction.
Thank you for your professional advice, i have deleted some detalis, and the topics selected in the paper are introduced in L70-79. and further introduction about four related species are moved to L154-162. And deleted the content in L90-93.
In addition, I recommend the authors to further correct and polish their manuscript, as there are still too many typos and inappropriate English expressions.
I am sorry for the many typoos and inappropriate English expressions. this manuscript has been polished by Englishlanguage company. and we also corrected and polished the manuscript again.